# Miniaturized Platform for Individual Coral Polyps Culture and Monitoring

**DOI:** 10.3390/mi11020127

**Published:** 2020-01-23

**Authors:** Yongsheng Luo, Jinglun Zhao, Chunpeng He, Zuhong Lu, Xiaolin Lu

**Affiliations:** 1State Key Laboratory of Bioelectronics, School of Biological Science & Medical Engineering, Southeast University, Nanjing 210096, China; 230189586@seu.edu.cn (Y.L.); cphe@seu.edu.cn (C.H.); 2School of Public Health, Nantong University, Nantong 226019, China; jlzhao_1992@163.com

**Keywords:** coral polyp, microfluidic, coral bleaching

## Abstract

Methodologies for coral polyps culture and real-time monitoring are important in investigating the effects of the global environmental changes on coral reefs and marine biology. However, the traditional cultivation method is limited in its ability to provide a rapid and dynamic microenvironment to effectively exchange the chemical substances and simulate the natural environment change. Here, an integrated microdevice with continuous perfusion and temperature-control in the microenvironment was fabricated for dynamic individual coral polyps culture. For a realistic mimicry of the marine ecological environment, we constructed the micro-well based microfluidics platform that created a fluid flow environment with a low shear rate and high substance transfer, and developed a sensitive temperature control system for the long-term culture of individual coral polyps. This miniaturized platform was applied to study the individual coral polyps in response to the temperature change for evaluating the coral death caused by El Nino. The experimental results demonstrated that the microfluidics platform could provide the necessary growth environment for coral polyps as expected so that in turn the biological activity of individual coral polyps can quickly be recovered. The separation between the algae and host polyp cells were observed in the high culture temperature range and the coral polyp metabolism was negatively affected. We believe that our culture platform for individual coral polyps can provide a reliable analytical approach for model and mechanism investigations of coral bleaching and reef conservation.

## 1. Introduction

Coral reefs are an important component of the marine ecosystem, where they have a great impact on maintaining global dynamic ecological balance [1,2,3,4,5]. The existential crisis of coral reefs has arisen, as the marine environment is changing, including ocean acidification and seawater temperature increase, resulted from the high carbon dioxide levels and El Nino phenomenon [6,7]. Especially for elevated seawater temperatures, which is driving the increase of coral bleaching and leading to the loss of coral cover in the whole world [8,9]. Normally, the prolonged coral bleaching over several months can lead to extensive coral death [10]. Therefore, understanding how elevated temperatures exacerbate the corals bleaching has significant implications for the coral reef conservation and recovery.

Corals are generally viewed as intracellular symbiotic systems with certain species of algae to supply oxygen and energy through photosynthesis [11]. The coral bleaching occurs when the symbiotic relationship is broken by external stresses, among which thermal stress is an important one. However, the bleaching mechanism due to the thermal stress effect is difficult to investigate because of the complex physiological connections to the metabolic process of the coral symbiotic systems [12,13,14]. One of the reasons for the fractured relationship may be that the zooxanthellae can produce harmful substances, e.g., reactive oxygen species under the high-temperature stress, which can directly inhibit the synthesis of D1 proteins that is necessary for the photosystem bioremediation [15]. Nesa et al. provided the experimental evidence for such remediation by exposing the tissue balls to thermal stress at the DNA and cellular levels [16]. Furthermore, the survival time of corals was found to be negatively correlated with the zooxanthella density, suggesting the loss of energy source was also an important factor [15,17]. Yanasivan et al. investigated the effect of thermal stress on whether it can be ameliorated by providing an alternative energy source in the symbiotic system. The result indicated that the feeding can improve the coral health and support the metabolic process during and after thermal stress [18]. However, current studies related to the effect of thermal stress on the coral bleaching mainly focus on the static culture of coral tissues or small branchs [3,13,19]. The accumulation of metabolic waste, consumption of dissolved oxygen (DO), and dissolved inorganic carbon (DIC) can cause the separation of the zooxanthellae from the coral host cells, and thus affect the reliability of experimental results [20]. This is a limiting factor for evaluating the effect of the environmental stress and may lead to failure of the experimental test. In addition, the individual differences of corals are another key factor to evaluate the effect of the environmental stress [3]. Therefore, there is a desperate need to develop a flexible integrated microdevice as an alternative to traditional methods to mimic the marine ecological environment. Based on this, the development of microfluidic device to achieve the dynamic culture and real-time monitoring on individual coral polyps in the controllable microenvironment offers a new methodology [21,22,23,24,25]. An open interface is provided in this microfluidic device to allow precise control for the culture environment. However, there are still some challenges yet to be overcomed related to this microdevice, such as the reliability, integration, and maneuverability [20,26,27,28,29].

Here, we develop a temperature-adjustable coral-polyp-on-chip device that provides sea-like culture conditions for the individual coral polyps. The dynamic culture platform and circulating water pipe heat transfer module are integrated to enable facilitated handling and a controllable microenvironment for the individual coral polyps. The flow field and sensitivity of the temperature-control module in the microfluidic device are optimized based on simulation analysis and experimental results. The potential for the long-term culture and monitoring of the individual coral polyps was also explored for this microwell-based microfluidic platform. The responses of the individual coral polyps under the different culture temperature conditions were analyzed, which can provide a new approach to investigate the coral bleaching mechanism.

## 2. Materials and Methods

### 2.1. Microdevice Fabrication

The miniaturized microfluidic culture platform consists of a number of different components, including culture chip for coral polyps, cover plate, and temperature controlling module. The module of culture chip and cover plate are used to provide a habitable environment for the coral polyps and set up a convenient platform for visualization of the coral dynamic culture at the mesoscopic scale. The substrates are made of the polymethylmethacrylate (PMMA) transparent materials with a length of 60 mm, width of 80 mm, and height of 9 mm. As shown in Figure 1a, the channels in the culture chip are engraved with a square width and a height of 1 mm × 300 μm. In the center of the substrate, three parallel microwells are created with a height of 500 μm and a diameter of 1.5 mm as separate cavities to accommodate the individual coral polyps. In addition, the barrier and buffer region are used to eliminate the impact of the shear force near each microwell, indicated by the dotted arrow in Figure 1b. The temperature controlling module is mainly composed of circular water pipes, engraved with a square width and a height of 4 mm × 2 mm on two plates. The two temperature controlling plates were integrated on the sidewall of the culture chip and cover plate, forming a “sandwich” structure to control and stablize the temperature when culturing the individual coral polyps. The temperature controlling plates need to be connected to a water circulation temperature controller (LX-150, Coolium Instruments, Beijing, China) via the silicon tube (diameter 6 mm).

The multilayer chips were fabricated via the micromachining technology. In summary, the template was designed by using the Software AutoCAD 2014 (Autodest, San Rafael, CA, USA) and processed with a computer-controlled milling machine (JD650, Jingdiaoweiye Co.,Ltd., Beijing, China). Before the experiment, all the layers were sequentially rinsed with ethanol, ultrapure water (18.2 MΩ, Milli-Q gradient A10 water purification system, Millipore Corporation, Burlington, MA, USA), and finally blown dry with N_2_.

### 2.2. Analysis of Flow Field and Heat Transfer

In this study, the inlet flow rate (*Q*) in the chip for culture of the coral polyps is less than 120 µL/min. The Reynolds number to characterize the flow can be given by:(1)$$Re=\frac{\rho QL}{\mu S}$$
where ρ is the fluid density, *S* is the cross-sectional area of the flow channel ($0.6{\;\mathrm{mm}}^{2}$), μ is the fluid viscosity, *L* is the characteristic dimension of the channel (300 μm). We assumed an incompressible, isothermal, Newtonian fluid with fluid density $\rho =1000\;\mathrm{kg}/m^{3}$, fluid viscosity μ=1 mPa·s, and the no-slip condition at the channel walls. When the Reynolds number is small enough, just like the one in our model (*Re* = 1), the velocity distribution and the shear stress in the steady-state can be obtained by solving the three-dimensional laminar Navier–Stokes equations.
(2)$$\rho \left(u\times \nabla \right)\times u=\nabla \times \left[-p+\mu \left(\nabla u+{\left(\nabla u\right)}^{T}\right)\right]$$
(3)ρ∇×(u)=0
where *u* is the velocity of the fluid and *p* is the local pressure.

The flow in the temperature controlling channels can be analyzed based on the momentum and mass conservation equations, which can also describe the nonisothermal pipe flow [30]:(4)$$\rho \frac{\partial u}{\partial t}=-\nabla p-\frac{1}{2}f_{D}\frac{\rho }{d_{h}}u\left|u\right|$$
(5)$$\frac{\partial A\rho }{\partial t}+\nabla \times \left(A\rho u\right)=0$$
where *A* is the cross-section area of the pipe, $f_{D}$ is the friction factor of flow. Based on the Churchill friction model, $f_{D}$ can be calculated by:(6)$$f_{D}=8{\left[{\left(\frac{8}{Re}\right)}^{12}+{\left(c_{A}+c_{B}\right)}^{-1.5}\right]}^{\frac{1}{12}}$$
where
(7)$$c_{A}={\left[-2.457\ln\left((\frac{7}{Re}{)}^{0.9}+0.27\left(\frac{e}{d_{h}}\right)\right)\right]}^{16}$$
(8)$$c_{B}={\left(\frac{37530}{Re}\right)}^{16}$$

As shown in the equations above, the friction factor depends on the surface roughness (*e*) and the cross-section length ($d_{h}$) of the pipe.

The wall heat transfer between the temperature controlling pipes and coral culture chip can be described by the energy equation:(9)$$\rho AC_{p}\frac{\partial T}{\partial t}+\rho AC_{p}u\nabla T=\nabla \cdot Ak\nabla T+\frac{1}{2}f_{D}\frac{\rho A}{d_{h}}\left|u\right|u^{2}+Q_{\mathrm{wall}}$$
here $C_{p}$ is the heat capacity at the constant pressure, *T* is the water temperature in the pipes, and *k* is the thermal conductivity. The fluid temperature in the coral culture chip is governed via the heat exchange with the surrounding PMMA block. This term $Q_{\mathrm{wall}}$ can be seen as a line heat source and coupled to the governing equation of solid heat transfer:(10)$$\rho C_{p}\frac{\partial T_{2}}{\partial t}=\nabla \cdot k\nabla T_{2}$$

The wall heat transfer term can be described by $Q_{\mathrm{wall}}=\left(hZ_{\mathrm{eff}}\right)\left(T_{\mathrm{ext}}-T\right)$, where *h*, $Z_{\mathrm{eff}}$, $T_{\mathrm{ext}}$ are the heat transfer coefficient, perimeter of the pipe and external temperature outside of the channel, respectively. In addition, the heat transfers between the PMMA block and external environment temperature can be calculated by the Equation (10) and the heat transfer term.

All the simulations were performed in the COMSOL Multiphysics 5.4 software (COMSOL Inc., Stockholm, Sweden) based on the finite element method. The parameters used in the model were summarized in Table 1.

### 2.3. Coral Polyps Bail-Out and Culture

In this study, the coral of *P. damicornis* (Coelenterata, Anthozoa, Scleractinia, collected from the coral reef in Wenchang, Hainan Province, China) was selected for a series of experiments. Based on the calcium-free cell dissociation protocol, the coral polyps were induced to bail out from the coral reefs [31]. First, the fast-growing apical fragments (length about 1.0 cm) of the coral reefs were excised from parent coral colonies and cut into smaller pieces (length about 0.3–0.5 cm) with a stainless scissor. Second, the small pieces were then immersed in the calcium-free artificial seawater (CaFSW) in a dish, washed three times, and pre-incubated in CaFSW with an orbital incubator (80 rpm) for 3 h. Then, they were transferred to a culture medium (about 5 mL) supplemented with the 3% penicillin-streptomycin solution in a 6 well-plate. The plate was placed in a 26 °C incubator and monitored continuously for the state of the coral polyps. Finally, the individual coral polyps were released from the fragmental skeleton after about 20 h, and the intact coral polyps were submerged in the filter-sterilized artificial seawater (FASW) for the initial recovery.

The culture mediums with a concentration of 20% (Dulbecco’s modified Eagle’s medium, DMEM) were prepared for bail-out of the coral polyp. The CaFSW was prepared by adding 23 g of NaCl, 0.763 g of KCl, 1.89 g of MgSO_4_·7H_2_O, 10.45 g of MgCl_2_·6H_2_O, 3 g of Na_2_SO_4_, and 0.25 g of NaHCO_3_ to a liter of deionized H_2_O. The pH value of the CaFSW was adjusted to be 8.0 using the NaOH solution. For the individual coral polyps culture, the culture process went with the salinity of 35 p.p.t (g/L), 12/12 h illumination/dark cycle (white/red/blue fluorescent bulbs), and pH of 8.0–8.2.

### 2.4. Temperature Measurement and Image Capture

The infrared radiation thermometer (Aicevoos, Shenzhen, China) was used to measure the temperature of the coral culture chip. The temperature of the microfluidic chip was measured as an average from three independent devices under the stable conditions. Imaging of the autofluorescenceswas based on the native green fluorescent proteins (green) in the *P. damicornis* coral polyps and algal chlorophyll (red) in the zooxanthellae [20]. The laser scanning confocal microscope (LSCM, Leica TCS SP8, Leica Microsystems Inc., Wetzlar, Germany) was used to capture images of the coral polyps and the fluorescent images were processed by the built-in software Leica Application Suite X.

## 3. Results

### 3.1. Optimization Design of Microfluidic Chip Structure

The fluid shear stress is a critical parameter under this study which directly affects the culture of the individual coral polyps [19,32]. When the microfluidic chip was designed, the shear rate value had to be used to evaluate the metabolic waste of the coral polyp, which can be taken away by the running fluid (seawater). In addition, the mass transfer also needs to be considered, which might be limited in a narrow space. Two consequences can thus be resulted in. One is the metabolic waste accumulation and the other is the low DO and DIC microenvironment, which affects the coral polyp culture a lot since the coral metabolism and photosynthesis are very sensitive to DO and DIC concentrations.

In order to obtain a suitable geometric structure and inlet flow rate, the flow characteristic was analyzed based on Equations (2) and (3). In this case, geometric parameters for the microwell, buffer region, and barrier were used to optimize the microfluidic chip design for culture of the individual coral polyp, as shown in Figure 2a. Figure 2b showed the flow distributions for models III–IV. As the microwell was added to the channel, a more uniform liquid flow at the center of the coral culture cavity was obtained. And, the design of the microwell avoids the directly flow impact during the coral polyp culture, which can be select as a core d feature. In addition, the flow field distribution was further optimized by the buffer region and barrier, as shown in Figure 2b (models III, IV). Judged from the simulation result, as shown in Figure 2c, the model IV presented a smoothest velocity distribution in the coral polyp culture cell, and the shear rate decreased by 80% compared to the initial model (I) after optimization. This can lead to a continuous steady flow in the cell which is conducive to the maintenance of the coral polyp morphology. However, the cost of low fluid shear is the decrease of the mean flow velocity, which negatively affects the mass transfer in the microfluidics. For the laminar flow with the low Reynolds number, the convection only transfers the substance by the tangent of the flow velocity (along streamline) and heavily relies on the mass diffusion. The effect of the flow velocity on mass transfer could be evaluated by the Peckert number:(11)$$Pe=\frac{LU}{D}$$
where *L*, *U*, *D* were the characteristic length (1.5 mm), flow velocity, and diffusion coefficient (for the oxygen $D=3\times 10^{-9}\;{\mathrm{mm}}^{2}/s$), respectively. From the simulation result of the model IV with various inlet rates (15, 30, 60, and 120 μL/min) (Figure 2e), we could calculate the dimensionless constant on the order of 0.5–20. The Peckert number greater than 1 indicates that the mass flux got the contribution from convection much more than diffusion [33,34]. Thus, the optimized inlet flow rate in this model was from 30 to 60 μL/min in order to reach a balance between the high mass transfer and low fluid shear stress, which would eventually be applied to the microwell. Moreover, after the optimization, the flow shear rate in the chip environment (1.5–2.5 s^−1^) is within the range of the natural environment (0.11–3 s^−1^) [35,36].

### 3.2. The Evaluation of the Temperature-Control Module

Temperature control is also an important step in the individual coral polyp culture. On one hand, failure to control the temperature may lead to the deterioration of the weatherability and the increase of the pathogen growth rates [37]. On the other hand, it has be reported that the environmental temperature changes can increase the thermal tolerance for the corals [38]. In this study, the temperature was controlled in the culture region by the heat exchange between the circulating water pipes and the culture chip.

The uniformity and stability of the temperature distribution in the coral polyp culture region should be considered when the circulating water pipes were designed. Additionally, in order to ensure monitoring and tracking of the coral polyp temperature, the circulating water pipes should not be mounted in this culture region. As a consequence, a modular detachable temperature-control setup was developed with an “nine-pipeline-format”. The heat transfer model is shown in Figure 3a, with the gray ones representing the distribution of the circulating water pipes and the blue ones (simulation probe region) representing the culture regions for the individual coral polyps. Therefore, using the conjugate heat transfer module, as shown in Figure 3b, a relatively uniform and stable surface temperature distribution map was obtained. As expected, the temperature on the surface of the microwell (~25 °C) was close to the setpoint (26 °C), although the temperature in the chip edge area was less than the anticipatory value.Further analysis indicated, as shown in Figure 3c, in less than 10 min the steady-state temperatures were acquired in the acceptable range of 24–26 °C under various environmental temperatures (5, 10, 15, 20, and 30 °C); while the optimally temperature range of 23–29 °C was known to be suitable for the growth of the reef-building corals [8,39]. The simulation results indicated that this circulating water pipe design could meet the requirement for the culture of the individual coral polyps. For example, as shown in Figure 3d, when the environmental temperature was 15 °C, the temperature in the probe region over the time was evaluated as a function of time with various set temperatures. However, it should be noted, the temperatures in the probe region were generally a little lower than the set values. This resulted from a tradeoff between the heat transfer and thermal dissipation. As shown in Figure 3e, the simulated temperatures with various set values (25, 26, 27, and 28 °C) in the steady state were also compared to the experimental probed ones. The results indicated that the real temperatures were a little lower than the simulated temperatures. This suggested that the set temperature for the circulating water pipes should be ~1.5 °C higher than the expected value.

### 3.3. Explantation of Individual Coral Polyps

Before culture in the microfluidic chip, the individual coral polyps needed to bail out from the coral reef skeleton via the calcium-free dissociation method and be carefully transplanted into the chip. Based on our previous experimental result, the coral coenosarc parts in the fast-growing apical fragments would go through a shrink stage, when the small fragments were submerged in the culture medium (Figure 4a).The intact coral polyps were then carefully collected after bailing-out from the skeleton.

The schematic diagram for explantation of individual coral polyps in the microfluidic chip was shown in Figure 4b. The coral polyps were transferred into the microwell with a pipette and clamped with the PMMA cover plate. The schematic of the 3D assembly was shown in Figure 4c. This reversible assembly method could easily accommodate and alleviate the coral polyps in the explantation process and then support recycling of the coral polyps. The whole device consisted of four main components (Figure 4d): (1) a microinjection pump, as a crucial part, to provide sufficient DO, DIC, and nutrients and remove waste metabolites; (2) a water circulation temperature controller for the multilayer chip; (3) a blue light-emitting diode (LED) to replenish the light; (4) a PMMA-based multilayer chip with microcavities and circulating water pipes for culture of the individual coral polyps. The multilayer chip was assembled by thin polydimethylsiloxane (PDMS) films (above 10 µm), as shown in Figure 4e. The interfaces of the culture and temperature-control module were connected though the polyether-ether-ketone (PEEK) connector and the silicone tube.

### 3.4. Evaluation of Culture of Individual Coral Polyps

In order to examine the reliability of this microfluidic device, the coral polyps were cultured with FASW under the continuous perfusion. The morphology of the coral polyp was monitored with LSCM, as shown in Figure 5. The excitation wavelength was 488 nm and the two receiving wavelength ranges were 500–550 nm (Green) and 600–650 nm (red). After the explantation, the coral polyps still remained the biological activity. In this study, the side and top views of the coral polyp were captured, as shown in Figure 5a,b. Furthermore, the motility of the coral polyp were increasing over the culture time. As shown in Figure 5c,d, revolving of the coral polyp was observed, after it was cultured for 3 days in the microfluidic chip. In particular, we found that the rotation speed of the coral polyp was strongly influenced by the light, which was higher in the bright than that in the dark environment.

The physiological relevance of the temperature stress to the individual coral polyps was then investigated. For comparison, the coral polyps were cultured both at 25 °C and a high temperature (31 °C). For the first day, there was no obvious morphological difference. However, at 31 °C, the zooxanthellae was found to detach from the coral tissueon the second day (Figure 6b) and the coral polyps eventually disintegrated within 3 days. At 25 °C, the coral polyp still retained its normal morphology on the second day (Figure 6a) and the third day, and most of the individual coral polyps could survive for above a week.

## 4. Discussion

Traditional culture of coral polyp was based on the replacement of the medium every 24 h in a biological incubator at a constant temperature [15,32,40,41]. Although the viable coral polyp was generally viewed as net autotrophic, some experiments have confirmed that the photosynthetic production rate by the algae well exceeds the respiration rates of the algae and coral [12,42]. One disadvantage of the static culture is the imbalance of the photosynthesis and respiration regarding the chemical substance [43]. The model presented here aims at designing three independent culture regions used to mimic the marine ecological environment. Additionally, the geometric construction and inlet flow rate of the coral culture region were optimized to achieve a low shear rate and high substance transfer rate based on the simulation analysis. Futhermore, the circulating water module was integrated into the microfluidic chip to control the temperature of whole component. From the evaluation result of the heat transfer, this microdevice can achieve culture of the coral polyp at various environmental temperatures and thus be employed to investigate the thermal stress effect on coral bleaching.

Based on previous studies, the high zooxanthellae density of the coral polyp could be seen as the manifestations of biological activity of the coral polyp [14,15,16]. In our individual coral polyp culture experiment, the rotation motility of the coral polyp was restored after the cultivation, which suggested that the coral polyp could repair the damage by performing its normal metabolic function. The microdevice could provide a necessary environment for the growth of the coral polyp as we expected. Furthermore, this microdevice platform can be used to study the thermal stress effect. The coral polyp gradually lost its biological activity at the high temperature (31 °C). Therefore, the bleaching mechanism might not only be resulted from the reactive oxygen species produced by zooxanthellae to damaged the coral host cells but also associated with DO, DIC, and nutrition uptaking [15,18,44]. However, in order to fully address this issue, a more in-depth investigation should be conducted in the near future, for example microsensors for detecting chemical substances can be integrated [45,46].

## 5. Conclusions

Overall, we had designed an integrated microdevice platform with the temperature control module for culture of the individual coral polyps. This platform can be feasibly applied to examine the environment variables by further integrating microsensors. The design and feasibility of this microdevice were evaluated by using the simulation analysis and the culture experiments. Our simulation results indicated that the design of the microwell-based microfluidic chip with the temperature control module could provide an excellent growth environment for culture of the coral polyps, including the uniform flow environment, rapid mass transfer, and precise temperature control. Therefore, the microdevice could facilitate the investigation of, such as nutrient substance uptake or the coral response to the environmental stressors. Moreover, this microdevice platform can potentially be employed to couple with other state-of-the-art techniques to provide quantitative analysis of the cellular behaviors [20,47].

## Figures and Tables

**Figure 1 micromachines-11-00127-f001:**
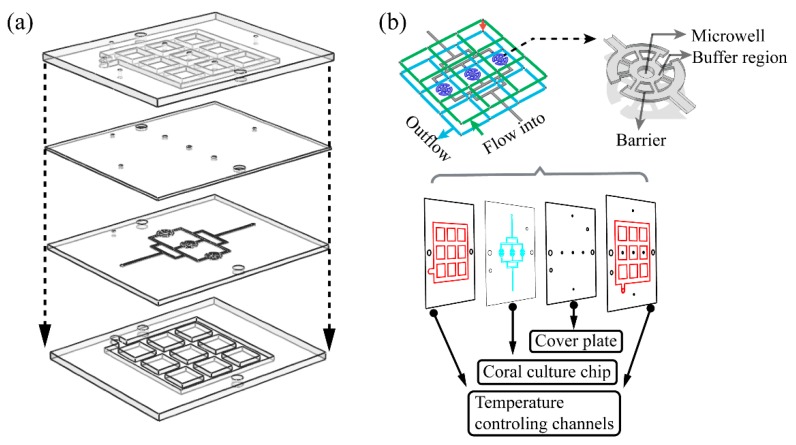
Schematics of the dynamic culture microdevice. (**a**) 3D assembly of different components. (**b**) Zoom-in images for the culture chip and temperature controlling module.

**Figure 2 micromachines-11-00127-f002:**
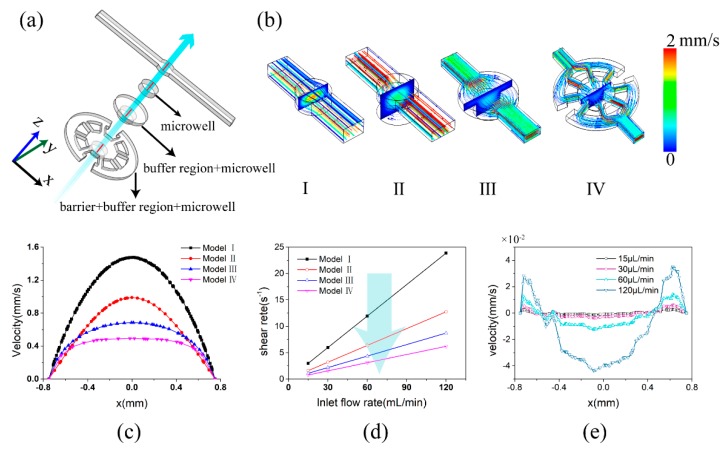
3D flow characteristics in the coral culture chip channel with a constant perfusion flow-rate. (**a**) The different design models (I, II, III, IV) for the coral culture. (**b**) The flow field profiles of the four models with the inlet flow rate of Q=30 μL/min. Flow velocity gradients were indicated by colors ranging from 0 (blue) to 2 mm/s (red). (**c**) The flow velocities along the *x*-axis passing through the center of the coral culture chip as the red lines marked in (**b**). (**d**) The mean shear rates in the red line regions at various simulation inlet flow rates for the four models. (**e**) The flow velocity profiles of the red line regions along the *z*-axis at various inlet flow rates for the model IV.

**Figure 3 micromachines-11-00127-f003:**
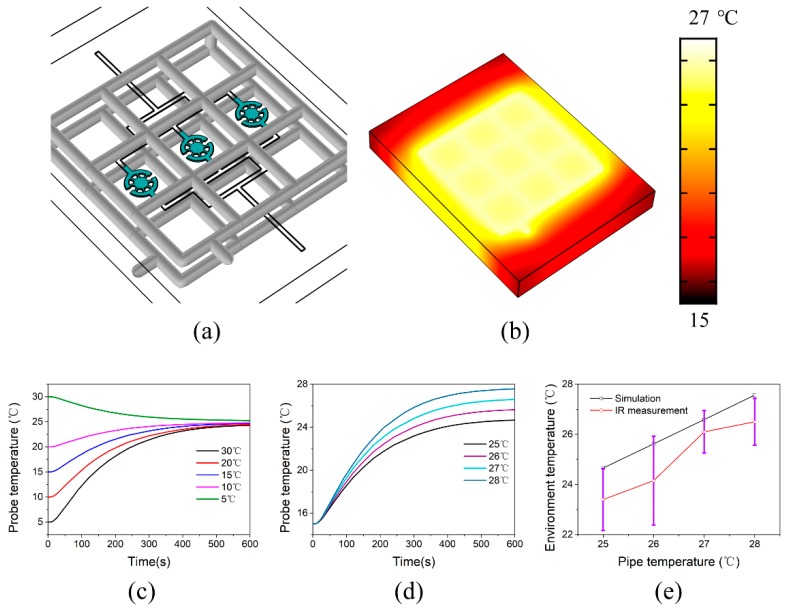
Ttemperature profiles of the microfluidic chip. (**a**) 3D geometric model of the temperature-control module. (**b**) The temperature map of the whole microfluidic chip in the steady-state. (**c**) Temperature changes in the probe region as a function of time with various environment temperatures. (**d**) Temperature changes as a function of time with various circulating water temperatures. (**e**) Comparison of surface temperatures in the probe region between simulated and IR measured ones in the steady-state (environment temperature ~15 °C). Three independent experiments were performed under the steady state condition. The steady state condition was assumed to be achieved when the temperature variation in the probe region was less than ~0.5 °C within 20 min.

**Figure 4 micromachines-11-00127-f004:**
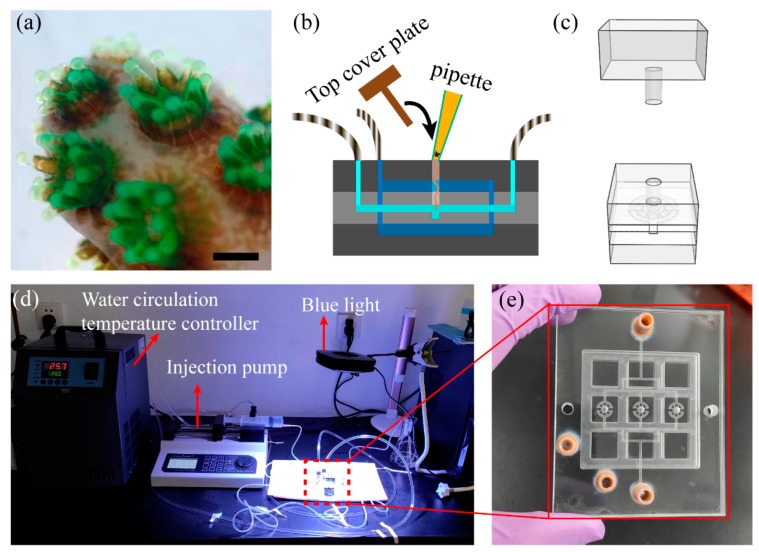
Explantation and culture of individual coral polyps. (**a**) Fluroscence images of the fast-growing apical fragments during the bail-out process (scale bar = 1 mm). (**b**) Schematic image illustrating the coral polyp explantation and assembly process. (**c**) Layout of the polyp culture cell. (**d**) Experimental setup for culturing individual coral polyps in the microfluidic device. (**e**) Photograph of a microfluidic chip.

**Figure 5 micromachines-11-00127-f005:**
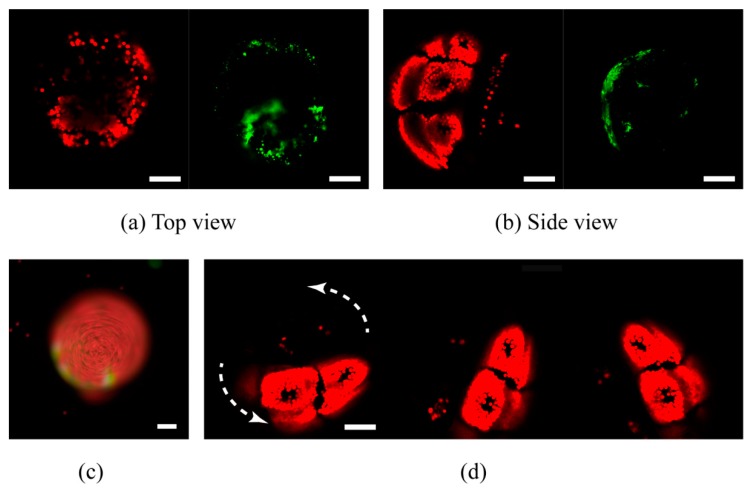
Confocal images of the individual coral polyp cultured in the microfluidic chip. (**a**) Top view and (**b**) side view. The autofluorescences came from the native green fluorescent proteins in the coral polyp (green) and algal chlorophyll (red). (**c**) Image of the coral polyp in the camera mode after culture in the microfluidic chip for 3 days (**d**) Confocal images confirmed the movement of the coral polyp in the microwell. All the scale bars are 100 µm.

**Figure 6 micromachines-11-00127-f006:**
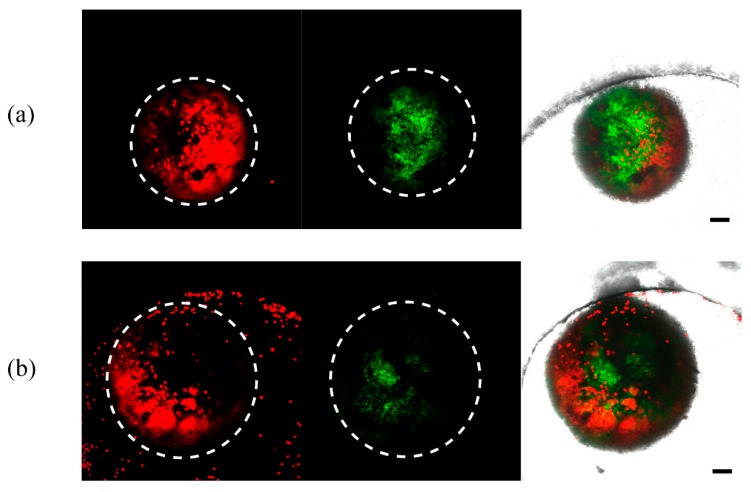
Impact of the temperature stress on the individual coral polyp. The morphologies of the coral polyps cultured at (**a**) 25 °C and (**b**) 31 °C in the microfluidic chip for 2 days. Left photographs are the autofluorescence images from the algal chlorophyll; middle photographs are the autofluorescences images from the native green fluorescent proteins; right photographs are the overlapped fluorescence images from both the algal chlorophyll and green fluorescent proteins (scale bar = 100 µm).

**Table 1 micromachines-11-00127-t001:** Parameters for the simulation model.

Parameters	Values
Density (PMMA)	1190 $\mathrm{kg}/m^{3}$
Heat capacity (PMMA)	1420 J/kg·K
Heat conductivity coefficient (PMMA)	0.19 W/m·K
Heat capacity (water)	4200 J/kg·K
Heat conductivity coefficient (water)	0.58 W/m·K
Surface roughness (e)	0.01 mm
The inlet flow rate of the pipe	10 L/min

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
