# Peer review of "Miniaturized Platform for Individual Coral Polyps Culture and Monitoring"

_micromachines, 2020, doi:10.3390/mi11020127_

Round 1
Reviewer 1 Report
In the manuscript, "Miniature platform for individual coral polyps culture and monitoring" by Luo and colleagues, the authors describe a microfabricated platform that provides environmental control to culture individual coral polyps on-chip, with the ability to dynamically modify inputs such as fluid flow and temperature.
Overall, a very nice manuscript that I feel will be of broad interest to the readers of Micromachines. From a grammatical standpoint, the English is very good, with only minor modifications needed. For example, on the last like of page one, "the bleaching mechanism due to the thermal stress effect is difficult to be investigated" should be "the bleaching mechanism due to the thermal stress effect is difficult to investigate".
The novelty of the paper is quite high, with the motivation for the research clearly outlined in the abstract. One question for the authors here. When discussing global warming and the effects of the coral caused by El Nino, the authors state that "the experimental results demonstrated that the microfluidics platform could provide the necessary growth environment for coral polyps as expected so that in turn the biological activity of the individual polyps can quickly be recovered." Given that they key parameters for polyp recovery can be identified on chip, how does this translate as a tool to the recovery of coral in the wild? Is there any hope that data from the chip can influence decisions that make a global impact in the future? Perhaps there is no good definitive answer to this question, but if there is, it would make the paper even stronger.
Under Section 2.1, Microdevice Fabrication, I would suggest discussing the device material used for the chip in the first paragraph. I realize that is is discussed in detail in paragraph 2 of the section, but even naming the substrate in paragraph one would make the narrative easier to read.
What computer-controlled milling machine was used? The stated roughness (0.01 mm) used in the estimations is very fine, so it would be good for readers to know what hardware was used if they want to duplicate the experiment.
Under Section 3.1, the authors discuss in detain the optimization of the flow conditions on-chip, with careful consideration of balancing the tradeoff between shear minimization and mass transport. One item in the discussion that is missing is the comparison to shear rates that the coral encounter in their natural environments. Please provide this data (even as a estimate), as it shows how flow in the chip environment compares to the ocean.
Under Section 3.2, the temperature control module, how tightly can the chip regulate temperature? As coral are very sensitive to even small changes in temperature, it would be good to report what level of thermal stability can be achieved on chip (0.1, 0.5?).
Under Section 3.3, last paragraph, the authors state that "a blue light-emitting diode (LED) to replenish the light". What is meant by "replenish the light"?
Under Section 3.4, what is meant by "After the explanation" in paragraph 1? Perhaps this is a grammatical error? Please revise.
The motivation
Reviewer 2 Report
Fig 3b. Is the map from one unit of the chip or the whole chip? Please describe clearly. Fig 3b. Is there any difference of the temperature measured at top- and bottom-side of the chip? Fig 3e. Why the error bars are big at 25 and 26C? How many repeats for each measurement? Please include in the figure legends. Fig 6. The GFP image should be also shown. It is not clear to see from the overlapped images. Fig 6. The separation of the zooxanthellae from the coral host cells should be quantitated. Coral is very sensitive to the temperature. 1 Celsius degree increase may separate zooxanthellae from the coral. In this work, only 25C and 31C were tested. The authors can test more temperatures in between and draw the correlation of separation between zooxanthellae from the coral. This result will demonstrate the power of this platform.Author Response
Please see the attachment.
